# Effect of Apple Cultivar and Selected Technological Treatments on the Quality of Apple Distillate

**DOI:** 10.3390/foods12244494

**Published:** 2023-12-15

**Authors:** Maria Balcerek, Katarzyna Pielech-Przybylska, Urszula Dziekońska-Kubczak, Anita Bartosik

**Affiliations:** Institute of Fermentation Technology and Microbiology, Faculty of Biotechnology and Food Sciences, Lodz University of Technology, 90-530 Lodz, Poland; katarzyna.pielech-przybylska@p.lodz.pl (K.P.-P.); urszula.dziekonska-kubczak@p.lodz.pl (U.D.-K.);

**Keywords:** apple cultivars, fermentation, distillation, volatile compounds, spirit, brandy

## Abstract

Apple producers are looking for new markets to dispose of their harvest surpluses. One of the solutions may be the production of apple spirits by small distilleries. This study aimed to evaluate the influence of apple cultivars and technological treatments, i.e., pasteurization, depectinization, and deacidification, on the fermentation efficiency and quality of the distillates. Samples for fermentation were prepared from Polish apple cultivars (Antonówka, Delikates, Kosztela, Kronselska). The control samples were raw pulp-based samples. After fermentation, the samples were analyzed for ethanol, residual sugars, and by-product content by the HPLC technique. The distillates were tested for volatile compounds by the GC-MS method and their sensory evaluation was performed. Raw pulp-based samples, independent of the apple cultivar, showed fermentation efficiencies between (75.77 ± 4.69)% and (81.36 ± 4.69)% of the theoretical yield. Depectinization of apple pulp prior to fermentation resulted in the highest ethanol concentration and yield up to approximately 89%. All tested apple distillates were rich in volatile aroma compounds and met the requirements of the EU regulation for hydrogen cyanide content. The obtained results indicate that the tested apple cultivars can be used for the efficient production of apple spirits, providing producers with an opportunity for brand development.

## 1. Introduction

In European countries, fruit distilleries are an important part of the economy and spirit drinks made from fruit distillates are known and appreciated worldwide. The tradition of fruit brandy production in Germany dates to the end of the 16th century and the Kirschwasser obtained from cherry spirit distillate is regarded as the most recognized German product [1]. In turn, France is world famous for cognacs and armagnacs; whereas, the Balkan countries are producers of slivovitz and rakija. Also, Poland has a long tradition of producing slivovitz, especially of Śliwowica Łącka produced in a sub-mountainous region of Poland [2,3].

The apple tree (*Malus domestica*) is the most frequently grown fruit tree in Poland, which is the largest producer of apples in the European Union (EU) and one of the largest producers in the world [4]. Poland’s share in the global production of apples is close to 4%; whereas, global exports reach 11% [5]. Apples are attractive fruits both for direct consumption and for the production of various types of apple products. Most of the apples processed in the country are mainly used for the production of apple juice concentrate (about 90%), fresh juices, smoothies, droughts, and alcoholic beverages [6].

The most popular apple alcoholic drink is cider with an alcohol strength by volume of 1.2–8.5% *v*/*v*. A much stronger drink made from distilled cider is calvados, whose name and production technology are reserved for the region of Normandy (France). An alternative to calvados in other countries is the production of apple spirit (brandy), of which the smell and taste are influenced by the variety of apples, as well as methods of production and aging [7].

Apple spirits can be obtained from the fermentation of both the fresh (raw) fruit pulp and juice, as well as from pomace, i.e., the solid residue obtained after apple processing [8]. A desirable liquefaction of the pulp, which has a beneficial effect on the course of fermentation, can be achieved by pretreating the pulp with pectolytic enzyme preparations. However, it should be noted that the use of pectolytic enzyme treatment may increase the methanol content in the distillate [9]. An alternative pretreatment of fruit pulp may be pasteurization, which also causes the liquefaction of the raw pulp, destroys the vegetative forms of microorganisms [10], and reduces the methanol content in the obtained apple distillates [11].

Among the factors determining the efficiency of fermentation and the quality of spirits is the pH of the fermentation medium. The results of the study by Liu et al. [12] showed that a low initial pH can cause a prolongation of the yeast lag phase. Moreover, it can affect the rate of consumption of total sugars, increase the final content of acetic acid and glycerol, and decrease the final content of ethanol. The application of chemical or microbial deacidification with the deacidifying yeast *Schizosacharomyces pombe* [13] of fruit pulp before fermentation can be helpful in eliminating the above-mentioned problems and can have a protective effect against corrosion of the equipment under the influence of acids [14]. On the other hand, it should be assessed whether this treatment does not cause changes in the quality of the distillate. For instance, Yang et al. [15] observed that the chemical deacidification of pear–kiwi fruit juice prior to fermentation caused a significant reduction in the content of aromatic compounds in the obtained wine.

Taking into consideration that apple growers are still looking for new markets for their harvest surpluses, one of the possible solutions could be the production of apple spirits by small distilleries on farms where apple orchards are located. The aim of this study was to evaluate the possibility of using old apple cultivars grown in Poland for spirit production. Moreover, the influence of selected technological treatments, i.e., pasteurization, depectinization, and deacidification on the fermentation efficiency and the quality of the obtained apple distillates was investigated.

## 2. Materials and Methods

### 2.1. Raw Materials, Microorganisms, and Supplements

Samples for fermentation were prepared from four Polish apple cultivars (Antonówka, Delikates, Kosztela, Kronselska) from an ecological orchard. The raw material was stored under refrigerated conditions until use. To prepare samples for fermentation, the apples were homogenized into pulp and subjected to one of the following treatments:-Pasteurization—samples of apple pulp were heated to 70–75 °C and kept at this temperature for 30 min;-Depectinization—a pectolytic enzyme preparation, ROHAPECT 10 L (AB Enzymes GmbH, Darmstadt, Germany), was added in the amount of 0.1 mL/kg of apple pulp and the pulp was heated at 50 °C for 30 min;-Deacidification—the pH of the apple pulp samples was adjusted with a sodium hydroxide solution (30% *w*/*w*) from natural value (approx. 3.5) to 5.0.

The appropriately prepared apple pulp batches were supplemented with Activit (Institut Oenologique de Champagne, Mardeuil, France) as a source of nutrients (i.e., a mixture of diammonique phosphate, inactive yeast, thiamine chlorhydrate) for yeast [16]. The preparation in the form of an aqueous suspension was dosed in the amount of 0.3 g/kg of fruit pulp.

Dry yeast Lalvin R2 (*S. bayanus*) (Lallemand Inc., Montreal, QC, Canada) in the amount of 0.3 g d.m./kg was used for the fermentation of the apple pulp.

### 2.2. Fermentation

The alcoholic fermentation of apple pulp was performed in 10 L glass bottles, each with 5 kg of apple pulp treated as described above and inoculated with hydrated yeast. The bottles were closed with fermentation locks containing paraffin oil, which allowed the release of carbon dioxide (CO_2_). Samples were kept at 18 ± 1 °C for 14–20 days. The samples were stirred occasionally and weight loss was measured daily. The process was continued until no further changes in the mass of the samples were observed.

### 2.3. Distillation

After the completion of fermentation, ethanol, and other volatile compounds, were distilled from the batches using a laboratory distillation unit, as described in our previous work [17]. Distillates containing 14–23% (*v*/*v*) ethanol were concentrated (without fractionation) to approximately 43 ± 1% (*v*/*v*) in a glass distillation apparatus equipped with a dephlegmator, according to Golodetz [18], and subjected to chemical analyses.

### 2.4. Analytical Methods

#### 2.4.1. Analysis of Apple Pulp before and after Fermentation

Apple pulp before fermentation was analyzed for:-Total extract, using a digital refractometer (Atago, Tokyo, Japan) calibrated in the weight percentage (% *w*/*w*) of sucrose [19];-Titratable acidity, expressed in grams of malic acid/100 g apple pulp. The assay involved diluting the portion of apple pulp (10 g) by adding 90 mL of distilled water and heating just to the boiling point. After cooling down, the solution was filtered and titrated with 0.1 M of NaOH solution against phenolphthalein [19]. The acidity of the apple pulp was calculated according to the following formula:
$$A=\frac{a\cdot n\cdot 0.067\cdot 100}{b}$$
where a is the volume of the 0.1 M NaOH solution [mL], n is the molality of the NaOH solution, 0.067 is the conversion factor to malic acid, and b is the weight of the apple pulp sample used for the assay;

-Sugars (glucose and fructose), by the HPLC technique [20]. To prepare apple pulp for sugar analysis, a sample weight of 10 g of apple pulp was centrifuged at 910× *g* for 10 min. The obtained supernatant was then clarified by using Carrez reagents and centrifuged again at the above-given conditions. The clarified solution was used for the determination of glucose and fructose. To determine the sucrose content in the apple pulp, the supernatant obtained after clarification was subjected to the inversion of sucrose by 36% *w*/*w* hydrochloric acid at 68–70 °C for 5 min. The solution after sucrose inversion was cooled to approx. 20 °C, neutralized with 25% (*w*/*w*) NaOH solution, and the content of the total sugars (i.e., the sum of glucose and fructose after the inversion of sucrose) was determined. Sucrose content was calculated by multiplying the difference between the sum of glucose and fructose after and before inversion, by the coefficient 0.95.

After the fermentation, in each sample, the concentration of ethanol, residual sugars (glucose, fructose, sucrose), and fermentation by-products (organic acids and glycerol) was analyzed by the HPLC method [20]. The samples for analysis were prepared analogously to the samples before fermentation.

#### 2.4.2. Analysis of Apple Distillates

An areometer calibrated for % of alcohol by volume (ABV) was used to determine the ethanol contents of the obtained distillates.

The analysis of apple distillates comprised the determination of acidity [21], volatile compounds [17], and free hydrocyanic acid [22]. Moreover, organoleptic assessment was carried out [23].

The acidity of the distillates was determined by the titrimetric method [21]. The assay involved diluting the tested portion of distillate with an equivalent amount of carbon-dioxide-free water and determining the acidity by titration with a standard sodium hydroxide solution against phenolphthalein. The acidity is expressed in gram acetic acid/L of alcohol 100% *v*/*v*.

The chromatographic analysis of volatile compounds in the obtained distillates was carried out using a gas chromatography (GC) apparatus (Agilent 7890 A, Santa Clara, CA, USA) with a mass spectrometer (Agilent MSD 5975C, Agilent Technologies, Santa Clara, CA, USA), as described by Pielech-Przybylska et al. [17]. A VF-WAX MS capillary column (60 m length, 0.50 μm film thickness, and 0.32 mm i.d.) was used to separate compounds. The GC oven temperature was programmed from 40 °C (6 min) to 80 °C at a rate of 2 °C/min and was then increased to 220 °C at a rate of 10 °C/min (hold time: 5 min). The flow rate of the carrier gas (helium) through the column was 2.0 mL/min. The temperature of the injector (split/splitless) was kept at 250 °C. Direct injections of tested distillates (1 μL) were made in split mode (1:40). Each sampling method was performed in triplicate. MS conditions were as follows: ion source temperature, 230 °C; transfer line temperature, 250 °C; and quadrupole temperature, 150 °C. The ionization energy was 70 eV.

Quantification of the volatile compounds was performed using calibration curves in the selected ion monitoring mode. Six calibration samples containing different concentrations of each standard compound were prepared. A 4-heptanone sample with a concentration of 45 mg/mL of alcohol 100% *v*/*v* in the analyzed samples was used as an internal standard to monitor the instrument response and retention time stability. Quantitative analysis was performed using Agilent MassHunter (USA) software(Version B.07.00/Build 7.0.457.0, Agilent Technologies, Inc. 2008, Santa Clara, CA, USA).

All gas chromatography standards were purchased from Sigma Aldrich (St Louis, MO, USA) and all were of GC purity. Standard solutions were prepared using anhydrous ethanol (Sigma-Aldrich) as a solvent and refrigerated at 4 °C for storage.

The content of free hydrocyanic acid (HCN) in the tested distillates was determined spectrophotometrically by the pyridine-pyrazolone method [22,23], using a cyanide test kit (Hach Company, Loveland, CO, USA). The method involved the conversion of HCN to cyanogen chloride using chloramine T solution. The cyanogen chloride was then reacted with a mixture of pyridine containing 1-phenyl-3-methyl-5-pyrazolone and 4.4-bis(1-phenyl-3-methyl-5-pyrazolone) to form a colored complex, the absorbance of which was measured spectrophotometrically at a wavelength of 490 nm. The concentration of hydrocyanic acid in the apple distillates was quantified using a standard curve prepared from NaCN solutions in the range of 0 to 1 mg of HCN equivalents/L of alcohol 100% *v*/*v*.

### 2.5. Statistical Analysis

All samples were prepared and analyzed in duplicate. The Statistica 10 software (TIBCO Software, Palo Alto, CA, USA) was used to provide the statistical analysis of the obtained results. Analysis of variance (ANOVA), followed by Tukey’s post hoc test, was used to evaluate the differences among the means at the significance level of 0.05.

## 3. Results and Discussion

### 3.1. Characteristics of the Tested Apple Cultivars

The contents of extract, sugars, and acidity are key parameters of fruits used in the production of fermented alcoholic beverages. The extract content in fruits is influenced by the conditions during the growing season, as well as the time of harvest and the storage period [24]. The extract of the analyzed apples ranged from (15.0 ± 0.5)% to (23.2 ± 0.7)% and depended on the apple cultivar. The highest content was found in the pulp of the Kosztela cultivar; meanwhile, cultivars such as Antonówka, Delikates, and Kronselska had a comparable content of the extract (Table 1).

Fruit acidity affects the sensory characteristics of dessert apples. Malic acid is one of the major acids present in apples. In our study, a significant influence of the cultivar on the content of organic acids in apples was demonstrated (Table 1). The highest acidity was found in the cultivar Antonówka (0.70 ± 0.04 g of malic acid/100 g of pulp) while the lowest was found in the cultivar Kosztela (0.16 ± 0.01 g of malic acid/100 g of pulp) (*p* < 0.05). The must of apple cultivars (Topaz, Eliza, and Rubin), tested by Tarko et al. [25], had an acidity of approx. 5–7 g of malic acid/L and the results of their study confirmed the influence of acidity on the course and efficiency of apple pulp fermentation. In addition, organic acids present in fruits can also influence the ester content of fruit distillates, which contributes to the pleasant fruity and floral aroma of fruit distillates [26]. Quality parameters (e.g., acidity, sugar content) of fruits depend on many factors. In addition to the apple cultivar, the location of the crop, the time of harvest, the ripeness, and the post-harvest treatment are of great importance [11].

Sugars are the most important component of fruit in terms of fermentation efficiency. The accumulation of sugars in fruit affects the alcohol content obtained during fermentation [27]. Fresh, ripe apples contain about 10–13% of the total sugars, with fructose dominating (5–7%) [28], which was confirmed by the results obtained in our study (Table 1). The highest concentration of reducing sugars (the sum of glucose and fructose), as well as sucrose, was found in the cultivar Koszela. The cultivars Delikates and Kronselska, on the other hand, contained lower levels of reducing sugars.

### 3.2. Influence of Apple Cultivar on the Results of Raw Pulp Fermentation

Alcoholic fermentation is a multi-stage anaerobic process carried out by microorganisms, such as yeast *Saccharomyces* cerevisiae, in which sugars are mainly converted into cellular energy and pyruvate generated from glucose metabolism is broken into ethyl alcohol and carbon (IV) dioxide [29]. As a result of apple pulp fermentation, a significant decrease in extract and sugar contents was observed in the post-fermentation batches compared to the samples before fermentation. On the other hand, the titratable acidity increased significantly, ranging from 1.73 ± 0.08 g of malic acid/100 g of apple pulp (Delikates) to 2.79 ± 0.22 malic acid/100 g of apple pulp (Kronselska) (Table 2).

The highest ethanol content (87.39 ± 5.42 g/kg) was determined after the fermentation of pulp based on the Kosztela cultivar while the batches of other apple cultivars showed similar alcohol concentrations, ranging from 55.97 ± 3.23 g/kg to 58.03 ± 2.14 g/kg (*p* > 0.05) (Table 3). However, when evaluating the ethanol yield in relation to the theoretical value calculated on the basis of the content of sugars present in fruit, all fermentation samples, regardless of the apple cultivar, showed similar values between (75.77 ± 4.69)% and (81.36 ± 4.69)% of the theoretical yield (*p* > 0.05). Januszek et al. [30] carried out a study on characteristics of fermented apple musts obtained from ten different apple cultivars (Elise, Rubin, Topaz, Golden Delicious, Szampion, Gloster, Pinova, Florina, Idared, and Jonagored), which were harvested in orchards of the Małopolska region (Poland). The authors observed that the fermentation dynamics of the musts varied depending on the used apple cultivars. For instance, the musts obtained from the cultivars Gloster and Florina fermented with higher dynamics than those obtained from other cultivars. As regards ethanol concentration determined in the musts after fermentation completion, it ranged from 4.1% *v*/*v* (Idared cultivar) to 6.5% *v*/*v* (Topaz cultivar). In turn, the lowest fermentation efficiency (61.3%) was observed for the samples from the cultivar Idared while the highest (94.7%) was observed for musts obtained from the cultivar Gloster.

During alcoholic fermentation, apart from ethanol, other by-products, such as glycerol and acids, are produced. The concentrations of these compounds in the tested fermentation samples varied according to the apple cultivar. Glycerol is a product of yeast metabolism and protects yeast cells from osmotic stress in a sugar-rich environment [31]. The main environmental factors influencing glycerol levels are sugar content, temperature, and pH [32]. Glycerol concentration after fermentation of the apple batches ranged from 2.20 ± 0.07 g/kg of pulp, based on the Antonówka cultivar, to 3.71 ± 0.11 g/kg of pulp, based on the Kosztela cultivar (*p* < 0.05).

All samples after fermentation contained non-volatile organic acid, i.e., malic acid, tartaric acid, succinic acid, lactic acid, formic acid, and acetic acid. Succinic acid is an intermediate product of the TCA cycle and one of the end products of the anaerobic metabolism of yeast [33]. In turn, the presence of lactic acid, formic acid, and acetic acid in the fermentation medium may be a result of the activity of homofermentative lactic acid bacteria [34]. The different concentrations of organic acids present in the fermentation samples may be associated both with their physico-chemical composition and the microbial ecosystem [35]. Fungi (yeasts and molds) and bacteria are naturally present on the surface of apples and can be found at each stage of the production of alcoholic beverages [36].

Among the studied apple cultivars, only the fermentation sample from the Koszela cultivar contained citric acid. It should also be noted that this sample contained the highest concentration of ethanol among the tested cultivars. This may explain the production of citric acid. Hauka et al. [37], who studied the effect of fermentation conditions on citric acid production from cassava peels, observed that an increase in the concentration of ethanol in the fermentation medium may be an agent that increases the production of citric acid. Moreover, the Kosztela-cultivar-based sample was characterized by the highest content of the majority of the determined by-products (apart from malic acid and formic acid). It is worth mentioning that Kosztela (Costel’s Apple) is one of the oldest Polish apple cultivars [38].

### 3.3. Influence of Apple Pulp Processing Method on Fermentation Results

The effect of the pretreatment of apple pulp from two cultivars, Kronselska and Delikates, on the fermentation results was assessed. The following types of pretreatment were used: (1) depectinization; (2) deacidification, consisting of adjusting the pH of the pulp from approx. 3.5 (natural pH) to 5 units; (3) pasteurization. A batch based on the raw apple pulp was used as a reference sample.

Both depectinization (particularly recommended during must pressing due to the hydrolysis of pectins) and pasteurization cause liquefaction of the fruit pulp, which may facilitate the fermentation process [14]. Deacidification, on the other hand, is a protective action against the corrosion of the equipment under the influence of acids. Moreover, the optimum pH for the activity of the yeast *S. cerevisiae* is thought to be in the range of 4–6 [12].

The application of different treatments to apple pulp before fermentation caused various changes in the physico-chemical composition of the fermentation batches.

It should be noted that the highest acidity (1.92 ± 0.07 g of malic acid/100 g of pulp) was found in the sample of pulp based on the Kronselska cultivar, which was deacidified before fermentation. Deacidification of the fruit pulp is useful in terms of protecting the equipment against corrosion and improving the quality of the stillage after distillation; however, on the other hand, this treatment increases the risk of microbiological contaminations, which may affect the quality and safety of the product [17]. This is likely to have happened in the mentioned fermentation batch. In turn, the fermentation batch prepared from the deacidified pulp of the Delikates cultivar showed an acidity value similar to that determined for the remaining fermentation samples of this cultivar. However, this sample contained a relatively high concentration of sugars, i.e., glucose and fructose, after fermentation. The fermentation of samples pretreated by pasteurization or depectinization did not show significantly different changes in the tested physico-chemical parameters compared to the sample of raw apple pulp (Table 4).

Concerning the effect of the pretreatments of the apple pulp on the ethanol production and the fermentation efficiency, the depectinization of the apple pulp of both tested cultivars, prior to the fermentation, resulted in the highest ethanol concentration and its yield in relation to the theoretical value (Table 5). On the other hand, the application of pasteurization did not lead to an increase in this parameter in relation to the fermentation samples prepared from the raw pulp. Deacidification of apple pulp was unfavorable from the point of view of ethanol yield. The fermentation batches of both apple cultivars showed a significantly lower ethanol yield (Table 5). It should be noted that deacidification of the fermentation medium provides a more suitable environment for both yeast [39] and for many bacterial species, including lactic acid bacteria [40]. The development of microbial contaminations may be a probable reason for the decrease in ethanol yield in the deacidified fermentation samples, especially from the Kronselska apple cultivar. The confirmation of the incomplete fermentation of the samples after deacidification is the relatively high content of residual fructose. The probable reason for the relatively high content of fructose in this sample may be a depletion of nutrients required for yeast to fully ferment this hexose sugar as a result of its utilization by the spoilage microflora present in the fermentation medium. Berthels et al. [41] observed that fructose utilization is more dependent on the supplementation of medium with assimilable nitrogen than glucose utilization. Moreover, the results of their study show that the discrepancy between glucose and fructose uptake during fermentation is not a fixed parameter but may depend on the yeast strain and the fermentation conditions. In addition, the samples of deacidified apple pulp after the completion of fermentation were characterized by a significantly higher content of lactic acid and acetic acid (Table 5). This can indicate the activity of spoilage microflora [42].

### 3.4. The Chemical Composition of the Obtained Distillates

The quality of the apple distillates obtained in our study was assessed by determining their total acidity by titration methods, hydrocyanic acid by the spectrophotometric method, and volatile compounds by the GC-MS-technique-based method.

#### 3.4.1. Acidity and Hydrocyanic Acid

The composition of the beverages depends mainly on the composition of the raw material. The fermentation and distillation processes also play a key role in the distilled drink [43,44].

The titratable acidity of tested distillates was closely related to the apple cultivar used. Among the obtained apple distillates, samples from the Kronselska and Antonówka cultivars showed the lowest acidity (between 156.4 ± 21.9 and 178.5 ± 24.9 mg of acetic acid/L of alcohol 100% *v*/*v*, *p* < 0.05) while the acidity of distillates from the Delikates and Kosztela cultivars was approximately two times higher (Figure 1).

The seeds of fruits, including apples, contain cyanogenic glycosides (mostly amygdalin), which are hydrolyzed to benzaldehyde and hydrogen cyanide [45]. As hydrocyanic acid is a toxic compound, its presence in foods and alcoholic beverages is strictly limited. According to Regulation (EU) 2019/787 of the European Parliament and of the Council [46], the hydrocyanic acid content in stone fruit spirits should not exceed 70 mg per liter of alcohol 100% *v*/*v* (i.e., 7 g/hL). All the tested apple distillates complied with this requirement; although, differences in concentration were observed depending on the cultivar. The lowest content of this compound was found in the distillates from the cultivars Antonówka and Kosztela (between 12.7 ± 1.9 and 15.9 ± 2.4 mg/L of alcohol 100% *v*/*v*, *p* < 0.05). The distillates from Delikates and Kronselska, on the other hand, contained hydrogen cyanide in approximately three times higher concentrations (Figure 1).

With regard to the effect of the apple pulp pretreatment on the acidity and the concentration of hydrocyanic acid, it was observed that the deacidification resulted in a significant increase in acidity of the distillates of both cultivars, probably due to the microbial infection, which was facilitated by adjusting the pH from approx. 3.5 to 5.

When comparing the effect of apple pulp pretreatment on the concentration of hydrocyanic acid in the obtained distillates, it can be seen that the highest amount of this compound was determined in the spirits obtained from the apple pulp of both cultivars pretreated by deacidification. This is because the optimal pH of around 5 encourages enzyme activity for catalyzing the hydrolysis of cyanogenic glycosides [47]. The application of depectinization to apple pulp also caused a significant increase in the concentration of hydrocyanic acid in the distillates compared to those from the raw pulp. This phenomenon may be due to the promiscuity of pectolytic enzymes catalyzing random side reactions in addition to their main reaction. On the other hand, the application of pasteurization to the apple pulp caused a decrease in the concentration of hydrocyanic acid in the distillates, from about 20% (distillate based on the Kronselska cultivar) to 36% (distillate based on the Delicates cultivar), in relation to the samples obtained from the raw pulp (Figure 2).

#### 3.4.2. Volatile Compounds Determined by GC-MS

From a quantitative point of view, the most important volatile compounds in fruit spirits are methanol and higher alcohols [48]. Methanol is formed during fermentation by the enzymatic hydrolysis of naturally occurring pectins. While methanol does not directly affect the flavor of the distillates, it is subjected to restrictive controls due to its high toxicity [49]. According to Regulation (EU) 2019/787 [46], the concentration of methanol in apple distillates should not exceed 1200 g per hectoliter of 100% *v*/*v* alcohol. The content of this compound varied considerably in the analyzed apple distillates. Its concentration depended both on the apple cultivar and on the method of pulp pretreatment prior to fermentation.

Among the tested apple cultivars, the highest concentration of methanol (1416.4 ± 42.49 mg/L of alcohol 100% *v*/*v*) was found in the distillate obtained after the fermentation of raw pulp based on the Kosztela cultivar while the lowest amount of this compound (935.6 ± 28.07 mg/L of alcohol 100% *v*/*v*) was determined in the distillate obtained after the fermentation of raw pulp based on the Antonówka cultivar (Table 6). In turn, the distillate obtained from depectinized apple pulp contained ten times higher concentrations of methanol than the samples obtained from raw pulp. For example, the concentration of this compound in the distillate from the depectinized pulp of the Delikates cultivar was 13,569.2 ± 407.08 mg/L of alcohol 100% *v*/*v* and in the distillate from the Kronselska cultivar, it was 14,891.1 ± 446.73 mg/L of alcohol 100% *v*/*v*. Moreover, due to the deacidification of the apple pulp, the concentration of methanol in the distillates was also high and amounted to 12,185.1 ± 365.55 mg/L of alcohol 100% *v*/*v* (from the Delikates cultivar) and 15,166.5 ± 455.00 mg/L of alcohol 100% *v*/*v* (from the Kronselska cultivar). The results indicate that pH is an important factor that significantly affects the activity of pectolytic enzymes. According to research conducted by Do Amaral et al. [50], pectin methylesterase has an optimum state at pH 8 and 50 °C while other authors indicated a pH of 5–6, and even in the range of 3.75 to 6, as optimal for pectin methylesterase [51]. Glatthar et al. [52] suggested a pH of 2.5 for fermentation in order to reduce the methanol content in the distillates. In addition, Chaiyasut et al. [53] suggested that thermal treatment of fruit pulp prior to fermentation at temperatures above 70 °C effectively prevents methanol production by inactivating pectin methylesterase. These assumptions were confirmed for distillates from the two tested apple cultivars (Delikates and Kronselska), whose pulp was pasteurized. The application of this pretreatment resulted in a lower methanol content in the distillates obtained, by approx. 60% for the distillate based on the Delikates cultivar and by 75% for the sample based on the Kronselska cultivar, compared to the concentration of this compound in the distillates obtained from raw pulp (*p* < 0.05) (Table 7).

In the distillates of agricultural origin, higher alcohols constitute the largest group of volatile compounds [54]. Active amyl alcohol (2-methyl-1-butanol), isoamyl alcohol (3-methyl-1-butanol), isobutyl alcohol (2-methyl-1-propanol), and 1-propanol are produced by yeast during alcoholic fermentation by the conversion of branched-chain amino acids (isoleucine, leucine, valine, and threonine) present in the medium [55]. Amyl alcohols are responsible for the flavor of many alcoholic beverages, such as whisky, and the quality of these beverages depends on their concentration [26]. All the obtained apple distillates were rich in higher alcohols. Among the identified and determined higher alcohols, the highest concentrations were observed for 3-methyl-1-butanol, 2-methyl-1-propanol, 2-methyl-1-butanol, 1-butanol, 1-propanol, and 1-hexanol. However, both apple cultivar and applied raw material pretreatment influenced their concentrations (*p* < 0.05). The reasons for the differences in the amounts of higher alcohols may be due to differences in the quantitative and qualitative composition of amino acids present in apple cultivars, sugar content, and the fermentation method [17].

Relatively small amounts of alcohols, including 2-propanol, 1-pentanol, 2-hexanol, furfuryl alcohol, and benzyl alcohol, were detected in the analyzed distillates (Table 6). Benzaldehyde and benzyl alcohol are formed during the hydrolysis of cyanogenic glycosides [45,47]. In turn, 1-hexanol is formed in plant tissues by the activity of the enzyme alcohol oxidoreductase on hexanal [56] and is considered to be an important contributor to the aroma of fresh fruit [57]. The level of 1-hexanol is considered to be of sensory relevance, particularly in the apple ciders, and is associated with a grassy scent in distillates. However, when it is present above 100 mg/L of alcohol 100% *v*/*v*, it causes a very intense grassy flavor and the distillates are unpleasant in both aroma and taste [58]. Concerning the effect of apple cultivar, the highest content of 1-hexanol (205.83 ± 6.17 mg/L of alcohol 100% *v*/*v*) was found in the distillate after fermentation of the raw pulp based on the Antonówka cultivar while the lowest (45.14 ± 1.35 mg/L of alcohol 100% *v*/*v*) was found in the distillate from raw pulp based on the Delikates cultivar. No strict correlation between the applied pretreatment of apple pulp and the content of this compound in the obtained distillates was observed (Table 7).

The tested apple distillates also contained an aromatic alcohol, 2-phenylethanol, which is synthesized by microorganisms using L-phenylalanine as substrate and contributes to the pleasant flavor of fruit distillates due to its very low odor threshold [59]. Distillates from different apple cultivars contained this compound in concentrations ranging from 69.08 ± 2.07 mg/L of alcohol 100% *v*/*v* (Delikates cultivar) to 176.58 ± 5.30 mg/L of alcohol 100% *v*/*v* (Kosztela cultivar) (Table 6). Regarding the applied pretreatments, significantly higher (*p* < 0.05) concentrations of 2-phenylethanol were found in the distillates obtained after the fermentation of pulp treated by depectinization than in the other apple distillates. The presence of alcohols, such as 1-nonanol and 1-octanol, was also determined in all distillates. Medina et al. [60] found these compounds in apple fruit, juice, and cider of different varieties (Festa, Branco, and Domingos). Additionally, 1-Octanol is a fatty alcohol that can also be produced by microorganisms using glucose as a substrate via a fatty acid synthesis pathway using various enzymes [61].

Aldehydes, including acetaldehyde, are among the volatile compounds found in distilled spirits and alcoholic beverages [62]. Acetaldehyde accounts for almost 90% of all carbonyl compounds in distillates. The organoleptic properties of acetaldehyde vary depending on its concentration. In general, aldehydes can be associated with green, grassy, and herbaceous notes [63] or an odor similar to that of overripe apples [64]. A high concentration of acetaldehyde, exceeding 125 mg/L, can negatively affect the sensory profile of spirits and other alcoholic beverages [65]. Depending on the apple cultivar used for fermentation, the concentration of acetaldehyde in the obtained distillates ranged from 72.63 ± 2.18 mg/L of alcohol 100% *v*/*v* (from the Kronselska cultivar) to 169.16 ± 5.07 mg/L of alcohol 100% *v*/*v* (from the Antonówka cultivar) (*p* < 0.05) (Table 6). The results of our study are in agreement with the findings of Januszek et al. [30], who determined similar concentrations of acetaldehyde (between 92 mg and 215 mg/L of alcohol 100% *v*/*v*) in the brandies obtained from different apple cultivars (Elise, Rubin, Topaz, Golden Delicious, Szampion, Gloster, Pinova, Florina, Idared, and Jonagored). Taking into consideration the effect of the pretreatment of the apple pulp on the acetaldehyde content, its concentrations were found to be significantly (*p* < 0.05) higher in the distillates obtained from the deacidified apple pulp than in the other distillates. Deacidification of apple pulp also caused at least a two-fold increase in the concentrations of 2,3-butanedione and acetone in the distillates compared to the samples obtained after the fermentation of both raw pulp and pulp after pasteurization or depectinization (Table 7). In addition to acetaldehyde, the relatively significant concentrations of aldehydes, such as isobutyraldehyde, nonanal, decanal, furfural, and benzaldehyde, were determined in the tested distillates. These compounds are characteristic of fruit spirits, including cognacs and calvados [66].

Other ubiquitous compounds in alcoholic beverages include the diketone 2,3-butanedione (diacetyl), which has a buttery aroma, as well as acetals, which are formed rapidly in distillates. The most prominent of the latter group is acetaldehyde diethyl acetal (1,1-diethoxyethane), a major flavoring component of distilled beverages, especially malt whisky and sherry [67,68]. This compound was present in very high concentrations in all tested apple distillates. Regarding the effect of apple cultivars, its concentrations in the distillates from the Delikates and Kronselska cultivars were similar and ranged from 526.7 ± 15.80 to 582.9 ± 17.49 mg/L of alcohol 100% *v*/*v* (*p* > 0.05); whereas, the distillates from the Antonówka and Kosztela cultivars contained more than double the concentrations of this acetal (Table 6). The application of pasteurization to the apple pulp of both tested cultivars (Delikates and Kronselska) resulted in an increase in the concentration of acetaldehyde diethyl acetal compared to the samples obtained from the raw pulp; whereas, depectinization caused the opposite changes, i.e., a decrease in the concentration of this compound (Table 7).

During alcoholic fermentation, many esters can be formed in the reaction between alcohols and acetyl-CoA, catalyzed by acetyltransferases and other enzymes. Esters are generally associated with a pleasant, fruity, and floral aroma. They are an important class of flavor compounds in distillates as they have low sensory thresholds; however, their contribution to flavor is strongly influenced by their concentration. The predominant ester synthesized by yeast is ethyl acetate, which is formed from ethanol and acetyl-CoA [69]. In our studies, its highest concentration was determined to be at the level of 1141.3 ± 23.1 mg/L of alcohol 100% *v*/*v* in the distillate obtained after the fermentation of the raw pulp of the Kosztela cultivar. The distillates from the Antonówka, Delikates, and Kronselska cultivars contained significantly lower (*p* < 0.05) amounts of this compound, ranging from 295.6 ± 14.8 to 397.8 ± 26.9 mg/L of alcohol 100% *v*/*v* (Table 6).

Taking into account the effect of the type of pretreatment of the apple pulp, a significantly higher concentration of ethyl acetate was found in the distillates obtained after fermentation of the deacidified pulp than in the other samples (*p* < 0.05). Contamination of the medium can be caused by bacterial activity at any stage of alcoholic fermentation. For example, acetic acid bacteria can produce large amounts of acetic acid and ethyl acetate, which are retained throughout the fermentation and can taint the final product. Although ethyl acetate production is increased at low oxygen levels, most of the ethyl acetate generated during acetic spoilage appears to be due to non-enzymatic esterification or the activity of other contaminating microorganisms [70]. Satora and Tuszyński [71] observed that the dominance of non-*Saccharomyces* yeast and other microorganisms during the turbulent phase of fruit must fermentation may contribute to the high levels of esters in the obtained distillates.

In addition to the dominant ethyl acetate, the tested apple distillates contained many other esters of acetic acid and higher alcohols, as well as esters of fatty acids and ethyl alcohol. Esters of acetic acid and higher alcohols, such as isoamyl (i.e., 3-methylbutyl) acetate, isobutyl acetate, and 2-phenylethyl acetate, are present in relatively significant amounts in all fruit distillates [72]. Among distillates from different apple cultivars, the relatively higher concentrations of isoamyl acetate were determined in those obtained from the Antonówka, Delikates, and Kronselska cultivars, rather than from the Kosztela cultivar. On the other hand, the samples obtained after the fermentation of the Kronselska and Kosztela cultivars contained significantly (*p* < 0.05) higher amounts of 2-phenylethyl acetate than the others (Table 6). The concentrations of these compounds were not strictly correlated with the pretreatment method applied to the apple pulp (Table 7).

Apple distillates contained, also, significant amounts of fatty acid esters, particularly ethyl octanoate, ethyl decanoate, and ethyl tetradecanoate, which contribute to the flavor of fruit distillates [73]. Their concentrations were also not strongly influenced by the type of apple pulp pretreatment.

The compound considered to give the distillates a buttery flavor and a rancid butter smell is ethyl lactate, with a perception threshold of 250 mg/L. Its presence may be the result of malolactic fermentation. However, if it is present in a concentration lower than approx. 150 mg/L [74], it is considered to be beneficial because it smoothens the firm character of alcoholic beverages [57]. Among the distillates obtained from different apple cultivars, a relatively high concentration of ethyl lactate was determined in the one obtained after fermentation of the raw pulp of the Kosztela cultivar. The application of the pasteurization or depectinization of apple pulp caused a decrease in the ethyl lactate concentration in the distillates compared to the samples obtained from the raw pulp (Table 7).

## 4. Conclusions

The results of this study indicate that the tested apple cultivars (Antonówka, Delikates, Kosztela, Kronselska) can be used for the efficient production of apple spirits. Moreover, the results show that the yield and quality of apple distillates are not only dependent on the apple cultivar but can also be modified by the pretreatment conditions of the raw material. This is supported by the observed changes in the chemical composition of the tested samples. Considering the increasing focus on niche craft products, old apple cultivars can serve as valuable raw materials for micro-distilleries, leading to unique distillates and spirit beverages—apple brandy with an interesting aroma and flavor—and providing producers with an opportunity for brand development.

## Figures and Tables

**Figure 1 foods-12-04494-f001:**
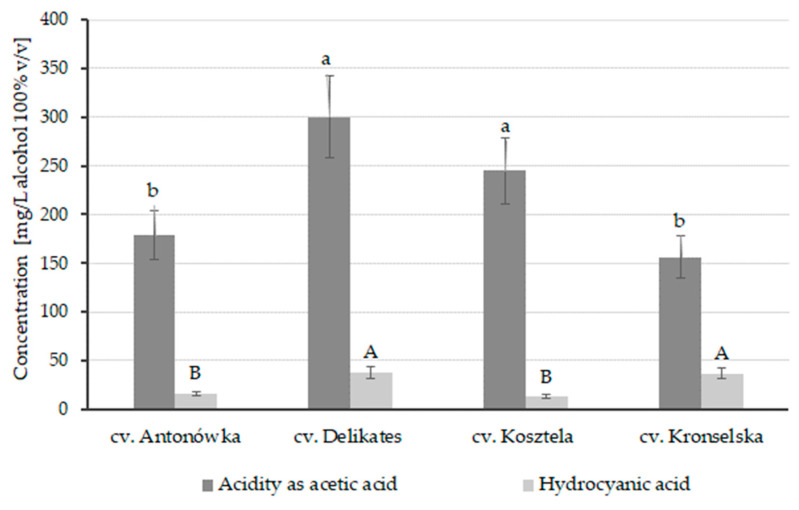
Acidity and hydrocyanic acid content in the distillates from the raw pulp of different apple cultivars. Different letters indicate significant differences (*p* < 0.05) between mean values of acidity (lowercase letters, a–b) and hydrocyanic acid (capital letters, A–B), as analyzed by one-way ANOVA and Tukey’s post hoc test.

**Figure 2 foods-12-04494-f002:**
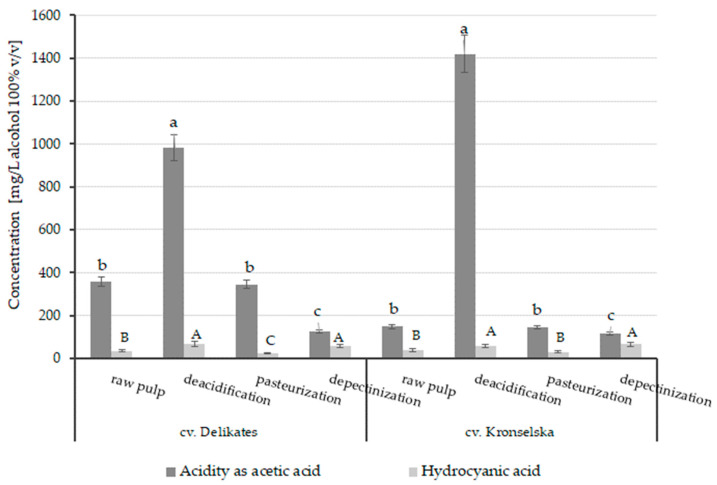
Effect of the pretreatment method of apple pulp on the acidity and hydrocyanic acid content in the distillates obtained. Different letters (separately for each cultivar) indicate significant differences (*p* < 0.05) between mean values of acidity (lowercase letters, a–c) and hydrocyanic acid (capital letters, A–C), as analyzed by one-way ANOVA and Tukey’s post hoc test.

**Table 1 foods-12-04494-t001:** Chemical composition of tested apple cultivars.

Apple Cultivar	Extract [% *w*/*w*]	Titratable Acidity [g Malic Acid/100 g]	Glucose [g/100 g]	Fructose [g/100 g]	Sucrose [g/100 g]
Antonówka	15.0 ± 0.5 ^b^	0.70 ± 0.04 ^a^	2.59 ± 0.30 ^b^	7.62 ± 0.44 ^ab^	4.22 ± 0.13 ^c^
Delikates	15.2 ± 0.4 ^b^	0.36 ± 0.02 ^c^	2.51 ± 0.27 ^b^	6.59 ± 0.41 ^b^	4.17 ± 0.13 ^c^
Kosztela	23.2 ± 0.7 ^a^	0.16 ± 0.01 ^d^	3.97 ± 0.36 ^a^	8.27 ± 0.68 ^a^	9.87 ± 0.29 ^a^
Kronselska	15.5 ± 0.3 ^b^	0.54 ± 0.03 ^b^	2.34 ± 0.28 ^b^	7.13 ± 0.46 ^ab^	4.97 ± 0.15 ^b^

Results expressed as mean values ± SD (n = 3); means in a column with different superscript letters (^a–d^) are significantly different (*p* < 0.05), as analyzed by one-way ANOVA and Tukey’s post hoc test.

**Table 2 foods-12-04494-t002:** Physico-chemical characteristics of raw pulp-based samples of different apple cultivars after fermentation completion.

Apple Cultivar	Extract [% *w*/*w*]	Titratable Acidity [g Malic Acid/100 g]	Glucose [g/100 g]	Fructose [g/100 g]	Sucrose [g/100 g]
Antonówka	3.4 ± 0.2 ^b^	2.51 ± 0.12 ^a^	0.14 ± 0.02 ^c^	0.34 ± 0.03 ^b^	0.15 ± 0.01 ^b^
Delikates	3.9 ± 0.1 ^a^	1.73 ± 0.08 ^b^	0.10 ± 0.01 ^d^	0.11 ± 0.01 ^c^	0.02 ± 0.01 ^c^
Kosztela	4.1 ± 0.2 ^a^	1.88 ± 0.05 ^b^	0.31 ± 0.02 ^b^	0.34 ± 0.02 ^b^	0.03 ± 0.01 ^c^
Kronselska	4.0 ± 0.1 ^a^	2.79 ± 0.22 ^a^	0.46 ± 0.02 ^a^	0.71 ± 0.03 ^a^	0.23 ± 0.01 ^a^

Results expressed as mean values ± SD (n = 3); means in a column with different superscript letters (^a–d^) are significantly different (*p* < 0.05), as analyzed by one-way ANOVA and Tukey’s post hoc test.

**Table 3 foods-12-04494-t003:** Composition of apple raw pulp-based batches after fermentation and efficiency of the process.

Compound	Antonówka	Delikates	Kosztela	Kronselska
Ethanol [g/kg]	58.03 ± 2.14 ^b^	55.97 ± 3.23 ^b^	87.39 ± 5.42 ^a^	57.29 ± 3.12 ^b^
Ethanol yield [% of the theoretical]	77.72 ± 2.87 ^a^	81.36 ± 4.69 ^a^	75.77 ± 4.69 ^a^	76.48 ± 4.16 ^a^
By-products [g/kg]				
Glycerol	2.20 ± 0.07 ^c^	3.02 ± 0.09 ^b^	3.71 ± 0.11 ^a^	3.07 ± 0.09 ^b^
Malic acid	0.01 ± 0.00 ^a^	0.01 ± 0.00 ^a^	0.01 ± 0.00 ^a^	0.01 ± 0.00 ^a^
Citric acid	n.d.	n.d.	0.39 ± 0.02	n.d.
Tartaric acid	0.09 ± 0.01 ^d^	0.18 ± 0.01 ^b^	0.26 ± 0.01 ^a^	0.14 ± 0.01 ^c^
Succinic acid	0.08 ± 0.01 ^b^	0.05 ± 0.01 ^c^	0.20 ± 0.01 ^a^	0.03 ±0.00 ^d^
Lactic acid	0.05 ± 0.00 ^d^	0.12 ± 0.01 ^c^	1.08 ± 0.03 ^a^	0.55 ± 0.02 ^b^
Formic acid	0.20 ± 0.03 ^b^	0.05 ± 0.01 ^c^	0.07 ± 0.01 ^c^	0.32 ± 0.03 ^a^
Acetic acid	0.12 ± 0.01 ^b^	0.20 ± 0.01 ^b^	0.34 ± 0.02 ^a^	0.10 ± 0.01 ^c^

Results expressed as mean values ± SD (n = 3); means in a row with different superscript letters (^a–d^) are significantly different (*p* < 0.05), as analyzed by one-way ANOVA and Tukey’s post hoc test; n.d.—not detected.

**Table 4 foods-12-04494-t004:** Effect of the pretreatment method of apple pulp samples on their physico-chemical composition after fermentation completion.

Cultivar	Pretreatment	Extract [% *w*/*w*]	Titratable Acidity [g Malic Acid/100 g]	Glucose [g/100 g]	Fructose [g/100 g]	Sucrose [g/100 g]
Delikates	Raw pulp	4.1 ± 0.4 ^b^	0.57 ± 0.02 ^a^	0.20 ± 0.01 ^b^	0.26 ± 0.02 ^b^	0.03 ± 0.00 ^b^
	Deacidification	5.5 ± 0.5 ^a^	0.51 ± 0.02 ^bc^	0.41 ± 0.06 ^a^	1.12 ± 0.06 ^a^	0.02 ± 0.00 ^b^
	Pasteurization	4.0 ± 0.2 ^b^	0.53 ± 0.03 ^ab^	0.09 ±0.01 ^c^	0.21 ± 0.01 ^b^	0.12 ± 0.00 ^a^
	Depectinization	4.3 ± 0.4 ^b^	0.47 ± 0.01 ^c^	0.15 ± 0.02 ^bc^	0.22 ± 0.03 ^b^	0.02 ± 0.00 ^b^
Kronselska	Raw pulp	4.2 ± 0.2 ^b^	0.65 ± 0.03 ^b^	0.13 ± 0.01 ^b^	0.36 ± 0.02 ^a^	0.07 ± 0.01 ^a^
	Deacidification	6.1 ± 0.5 ^a^	1.92 ± 0.07 ^a^	0.13 ± 0.01 ^b^	0.17 ± 0.02 ^c^	0.04 ± 0.01 ^b^
	Pasteurization	4.0 ± 0.2 ^b^	0.64 ± 0.03 ^b^	0.12 ± 0.01 ^b^	0.25 ± 0.02 ^b^	0.03 ± 0.01 ^b^
	Depectinization	4.2 ± 0.2 ^b^	0.67 ± 0.03 ^b^	0.18 ± 0.01 ^a^	0.26 ± 0.02 ^b^	0.02 ± 0.00 ^c^

Results expressed as mean values ± SD (n = 3); means in a column with different superscript letters (^a–c^), separately for each apple cultivar, are significantly different (*p* < 0.05), as analyzed by one-way ANOVA and Tukey’s post hoc test.

**Table 5 foods-12-04494-t005:** Effect of apple pulp pretreatment on the qualitative and quantitative composition of the samples after fermentation completion and ethanol yield.

Compound [g/kg]	cv. Delikates				cv. Kronselska			
	Raw Pulp	Deacidification	Pasteurization	Depectinization	Raw pulp	Deacidification	Pasteurization	Depectinization
Ethanol	53.33 ± 2.18 ^b^	35.33 ± 1.76 ^c^	55.59 ± 1.23 ^b^	61.28 ± 1.93 ^a^	56.30 ± 3.79 ^b^	30.57 ± 1.62 ^c^	55.73 ± 0.77 ^b^	63.70 ± 0.83 ^a^
Ethanol yield [% of theoretical]	77.52 ± 3.17 ^b^	51.36 ± 2.56 ^c^	80.81 ± 1.79 ^b^	88.57 ± 2.00 ^a^	75.16 ± 5.06 ^b^	40.81 ± 2.16 ^c^	74.39 ± 1.03 ^b^	85.03 ± 1.11 ^a^
By-products [g/kg]								
Glycerol	3.84 ± 0.12 ^b^	3.32 ± 0.09 ^c^	3.17 ± 0.09 ^c^	4.23 ± 0.13 ^a^	2.84 ± 0.09 ^b^	2.65 ± 0.08 ^c^	2.29 ± 0.07 ^d^	5.03 ± 0.15 ^a^
Glucose	1.26 ± 0.03 ^b^	1.49 ± 0.04 ^a^	0.64 ± 0.02 ^c^	0.15 ± 0.01 ^d^	0.53 ± 0.02 ^a^	0.48 ± 0.01 ^b^	0.46 ± 0.01 ^b^	0.18 ± 0.01 ^c^
Fructose	3.37 ± 0.10 ^c^	6.61 ^a^ ± 0.41	2.37 ^d^ ± 0.07	5.15 ^b^ ± 0.15	2.76 ^d^ ± 0.08	7.83 ^a^ ± 0.03	3.20 ^c^ ± 0.09	4.62 ^b^ ± 0.14
Arabinose	n.d.	n.d.	n.d.	0.36 ± 0.01	n.d.	n.d.	n.d.	n.d.
Malic acid	n.d.	1.01 ± 0.03	n.d.	n.d.	n.d.	0.49± 0.01	n.d.	n.d.
Citric acid	0.25 ± 0.01 ^b^	0.35 ± 0.01 ^a^	n.d.	n.d.	n.d.	0.34 ± 0.01 ^a^	n.d.	0.16 ± 0.01 ^c^
Tartaric acid	0.13 ± 0.01 ^c^	0.76 ± 0.02 ^b^	0.10 ± 0.01 ^d^	1.77 ± 0.05 ^a^	0.19 ± 0.01 ^c^	0.70 ± 0.03 ^b^	n.d.	1.16 ± 0.03 ^a^
Succinic acid	0.49 ± 0.02 ^a^	0.18 ± 0.01 ^b^	0.52 ± 0.02 ^a^	n.d.	0.59 ± 0.02 ^c^	0.66 ± 0.02 ^b^	0.41 ± 0.01 ^d^	0.76 ± 0.02 ^a^
Lactic acid	1.07 ± 0.03 ^a^	1.14 ± 0.04 ^a^	0.17 ± 0.02 ^b^	0.13 ± 0.02 ^b^	1.79 ± 0.05 ^b^	3.13 ± 0.06 ^a^	0.06 ± 0.00 ^c^	n.d.
Acetic acid	3.68 ± 0.11 ^a^	1.95 ± 0.06 ^c^	2.39 ± 0.07 ^b^	0.51 ± 0.02 ^d^	3.20 ± 0.10 ^b^	15.14 ± 0.45 ^a^	1.20 ± 0.03 ^c^	1.07 ± 0.03 ^d^

Results expressed as mean values ± SD (n = 3); means in a row with different superscript letters (^a–d^), separately for each apple cultivar, are significantly different (*p* < 0.05), as analyzed by one-way ANOVA and Tukey’s post hoc test; n.d.—not detected.

**Table 6 foods-12-04494-t006:** The effect of apple cultivar on the concentrations of volatile compounds in the distillates obtained.

Volatile Compound [mg/L of alcohol 100% *v*/*v*]	cv. Antonówka	cv. Delikates	cv. Kosztela	cv. Kronselska
Methanol	935.60 ± 28.07 ^a^	1061.90 ± 31.86 ^b^	1416.40 ± 42.49 ^c^	1084.60 ± 32.54 ^b^
2-Propanol	3.33 ± 0.10 ^b^	6.35 ± 0.19 ^c^	8.44 ± 0.25 ^d^	0.99 ± 0.03 ^a^
1-Propanol	251.10 ± 7.53 ^a^	232.40 ±6.97 ^a^	394.30 ± 11.83 ^c^	290.20 ± 8.71 ^b^
2-Methyl-1-propanol	1152.20 ± 34.57 ^c^	1052.60 ± 31.58 ^b^	776.90 ± 23.31 ^a^	1110.60 ± 33.32 ^bc^
1-Butanol	450.70 ± 13.52 ^c^	86.40 ± 2.59 ^a^	167.70 ± 5.03 ^b^	470.10 ± 14.10 ^c^
2-Methyl-1-butanol	859.00 ± 25.77 ^c^	707.60 ± 21.23 ^a^	891.20 ± 26.74 ^c^	779.00 ± 23.37 ^b^
3-Methyl-1-butanol	3786.70 ± 113.60 ^c^	2685.80 ± 80.57 ^a^	3098.00 ± 92.94 ^b^	2820.50 ± 84.61 ^a^
1-Pentanol	8.53 ± 0.26 ^d^	4.60 ± 0.14 ^b^	6.86 ± 0.21 ^c^	3.99 ± 0.12 ^a^
2-Hexanol	4.75 ± 0.14	n.d.	n.d.	n.d.
1-Hexanol	205.83 ± 6.17 ^d^	45.14 ± 1.35 ^a^	89.97 ± 2.70 ^b^	171.42 ± 5.14 ^c^
Furfuryl alcohol	3.15 ± 0.09 ^b^	2.47 ± 0.07 ^a^	4.06 ± 0.12 ^c^	7.21 ± 0.22 ^d^
Benzyl alcohol	17.68 ± 0.53 ^b^	14.47 ± 0.43 ^a^	23.47 ± 0.70 ^c^	16.57 ± 0.50 ^b^
2-Phenyletanol	105.33 ± 3.16 ^c^	69.08 ± 2.07 ^a^	176.58 ± 5.30 ^d^	84.43 ± 2.53 ^b^
1-Nonanol	37.98 ± 1.14 ^b^	45.47 ± 1.36 ^c^	34.47 ± 1.03 ^a^	47.03 ± 1.41 ^c^
2-Butanol	0.53 ± 0.02 ^d^	0.25 ± 0.01 ^a^	0.46 ± 0.01 ^c^	0.29 ± 0.01 ^b^
1-Octanol	25.19 ± 0.76 ^b^	20.63 ± 0.62 ^a^	33.45 ± 1.12 ^c^	23.62 ± 0.71 ^b^
Acetaldehyde	169.16 ± 5.07 ^d^	91.32 ± 2.74 ^b^	146.30 ± 4.39 ^c^	72.63 ± 2.18 ^a^
Isovaleraldehyde	2.30 ± 0.07 ^d^	0.94 ± 0.03 ^b^	1.99 ± 0.06 ^c^	0.67 ± 0.02 ^a^
2-Methylbutyraldehyde	1.01 ± 0.05 ^c^	0.87 ± 0.03 ^b^	1.08 ± 0.07 ^c^	0.44 ± 0.01 ^a^
Valeraldehyde	0.26 ± 0.01 ^b^	0.43 ± 0.01 ^d^	0.35 ± 0.01 ^c^	0.17 ± 0.01 ^a^
Heksanal	6.09 ± 0.48 ^b^	3.65 ± 0.21 ^a^	n.d.	3.20 ± 0.24 ^a^
Phenylacetaldehyde	1.10 ± 0.13 ^c^	0.84 ± 0.03 ^b^	1.26 ± 0.14 ^c^	0.58 ± 0.02 ^a^
Nonanal	26.34 ± 1.79 ^b^	14.15 ± 0.92 ^a^	12.72 ± 0.88 ^a^	14.49 ± 0.83 ^a^
Decanal	11.36 ± 0.34 ^b^	9.35 ± 0.28 ^a^	13.81 ± 0.41 ^c^	9.52 ± 0.29 ^a^
Furfural	111.10 ± 3.33 ^b^	18.26 ± 0.55 ^a^	17.92 ± 0.54 ^a^	20.82 ± 0.62 ^a^
Benzaldehyde	8.15 ± 0.24 ^a^	11.12 ± 0.33 ^b^	11.23 ± 0.34 ^b^	12.72 ± 0.38 ^c^
Isobutyraldehyde	44.17 ± 1.33 ^a^	44.30 ± 1.33 ^a^	n.d.	43.16 ± 1.29 ^a^
Trans-2-heptanal	18.24 ± 0.55 ^a^	n.d.	n.d.	19.10 ± 0.57 ^a^
3-Etoksypropionaldehyde	16.60 ± 0.50 ^a^	n.d.	n.d.	16.71 ± 0.50 ^a^
2-Propanone (acetone)	5.50 ± 0.17 ^b^	7.79 ± 0.23 ^d^	4.88 ± 0.15 ^a^	7.07 ± 0.21 ^c^
2,3-Butanedione (diacetyl)	3.63 ± 0.11 ^a^	3.57 ± 0.11 ^a^	3.72 ± 0.11 ^a^	3.03 ± 0.09 ^b^
Acetaldehyde diethyl acetal	1355.10 ± 40.65 ^b^	526.70 ± 15.80 ^a^	1310.60 ± 39.32 ^b^	582.90 ± 17.49 ^a^
2.3-Pentanodione	n.d.	10.58 ± 0.32 ^a^	n.d.	8.51 ± 0.26 ^a^
3-Octanone	25.18 ± 0.76 ^a^	27.90 ± 2.84 ^a^	28.43 ± 2.85 ^a^	29.36 ± 1.88 ^a^
Ethyl formate	n.d.	n.d.	1.05 ± 0.05	n.d.
Ethyl acetate	397.80 ± 26.90 ^b^	295.60 ± 14.80 ^a^	1141.20 ± 23.10 ^c^	390.60 ± 17.50 ^b^
Isobutyl acetate	0.26 ± 0.11 ^a^	0.13 ± 0.05 ^a^	0.33 ± 0.08 ^a^	0.19 ± 0.05 ^a^
Butyl acetate	1.08 ±0.05 ^b^	n.d.	4.88 ± 0.24 ^c^	0.14 ± 0.01 ^a^
3-Methylbutyl acetate	16.47 ± 0.32 ^c^	11.11 ± 0.25 ^b^	2.78 ± 0.14 ^a^	11.57 ± 0.28 ^b^
2-Methylbutyl acetate	1.09 ± 0.03 ^a^	1.12 ± 0.03 ^a^	1.28 ± 0.06 ^b^	1.09 ± 0.05 ^a^
Hexyl acetate	0.16 ± 0.03 ^a^	n.d.	0.11 ± 0.04 ^a^	0.14 ± 0.04 ^a^
2-Phenylethyl acetate	3.41 ± 0.18 ^a^	3.26 ± 0.07 ^a^	5.69 ± 0.23 ^b^	8.20 ± 0.04 ^c^
Methyl acetate	2.41 ± 0.18 ^a^	2.15 ± 0.35 ^a^	3.48 ± 0.23 ^b^	11.65 ± 0.75 ^c^
Ethyl propanoate	2.28 ± 0.11 ^b^	0.94 ± 0.18 ^a^	2.05 ± 0.13 ^b^	1.32 ± 0.27 ^a^
Isoamyl propanoate	0.07 ± 0.04	n.d.	n.d.	n.d.
Ethyl isobutyrate	0.10 ± 0.03 ^a^	0.13 ± 0.05 ^a^	0.15 ± 0.04 ^a^	0.11 ± 0.03 ^a^
Ethyl butyrate	0.95 ± 0.09 ^a^	0.81 ± 0.14 ^a^	1.28 ± 0.36 ^a^	1.08 ± 0.15 ^a^
Ethyl 2-methylbutyrate	0.15 ± 0.04 ^a^	0.24 ± 0.06 ^a^	0.72 ± 0.14 ^b^	0.22 ± 0.08 ^a^
Ethyl 3-methylbutyrate	n.d.	0.07 ± 0.04 ^a^	0.03 ± 0.02 ^a^	n.d.
Ethyl hexanoate	2.78 ± 0.14 ^a^	2.98 ± 0.15 ^a^	2.99 ± 0.15 ^a^	5.35 ± 0.27 ^b^
Ethyl heptanoate	0.09 ± 0.03 ^a^	0.10 ± 0.01 ^a^	0.10 ± 0.01 ^a^	0.15 ± 0.06 ^a^
Ethyl octanoate	16.96 ± 1.85 ^ab^	13.70 ± 1.69 ^a^	20.14 ± 1.15 ^b^	14.07 ± 1.70 ^a^
Ethyl nonanoate	0.54 ± 0.06 ^a^	0.41 ± 0.09 ^a^	0.53 ± 0.08 ^a^	0.45 ± 0.07 ^a^
Ethyl decanoate	25.40 ± 2.27 ^b^	31.65 ± 2.58 ^bc^	38.89 ± 4.94 ^c^	16.52 ± 1.83 ^a^
Ethyl lactate	2.37 ± 0.42 ^a^	1.62 ± 0.48 ^a^	61.92 ± 3.10 ^c^	14.32 ± 0.72 ^b^
Ethyl benzoate	21.38 ± 1.07 ^b^	13.75 ± 0.69 ^a^	14.20 ± 0.71 ^a^	13.27 ± 0.66 ^a^
Ethyl tetradecanoate	6.18 ± 0.81 ^ab^	14.08 ± 2.57 ^c^	10.54 ± 2.53 ^bc^	5.29 ± 0.86 ^a^

Results expressed as mean values ± SD (n = 3); means in a row with different superscript letters (^a–d^) are significantly different, as analyzed by one-way ANOVA and Tukey’s post hoc. test; n.d.—not detected.

**Table 7 foods-12-04494-t007:** The effect of apple pulp pretreatment on the concentrations of volatile compounds in the distillates obtained.

**Compound** **[mg/L of alcohol 100% *v*/*v*]**	**Delikates**				**Kronselska**			
	**Raw Pulp**	**Deacidification**	**Pasteurization**	**Depectinization**	**Raw Pulp**	**Deacidification**	**Pasteurization**	**Depectinization**
Methanol	1231.80 ± 36.95 ^c^	12,185.10 ± 365.55 ^b^	495.90 ± 35.85 ^d^	13,569.20 ± 407.108 ^a^	1412.10 ± 42.36 ^b^	15,166.50 ± 455.00 ^a^	339.70 ± 10.19 ^b^	14,891.10 ± 446.73 ^a^
2-Propanol	3.48 ±0.42 ^a^	n.d.	1.14 ± 0.23 ^b^	2.86 ±0.29 ^a^	n.d.	2.75 ± 0.08 ^a^	n.d.	n.d.
1-Propanol	264.20 ± 17.93 ^a^	271.80 ± 18.15 ^a^	198.50 ± 15.96 ^b^	144.70 ± 14.34 ^c^	385.40 ± 21.56 ^bc^	439.80 ± 33.19 ^a^	394.50 ± 21.84 ^bc^	368.50 ± 25.06 ^c^
2-Methyl-1-propanol	894.50 ± 26.84 ^b^	1127.70 ± 33.83 ^a^	746.40 ± 22.39 ^c^	718.10 ±21.54 ^c^	592.90 ± 17.79 ^d^	1038.70 ± 42.19 ^a^	678.20 ± 20.35 ^b^	747.30 ± 22.42 ^b^
1-Butanol	116.00 ± 5.50 ^a^	145.90 ± 47.40 ^a^	87.80 ± 26.60 ^a^	97.60 ± 25.90 ^a^	376.50 ± 11.30 ^b^	369.30 ± 11.10 ^b^	233.10 ± 6.90 ^c^	596.80 ± 17.90 ^a^
2-Methyl-1-butanol	575.60 ± 17.30 ^c^	685.16 ± 20.56 ^ab^	643.80 ± 19.30 ^b^	714.80 ± 21.40 ^a^	677.10 ± 20.30 ^b^	1107.60 ± 33.21 ^a^	549.60 ± 16.50 ^c^	583.60 ± 17.52 ^c^
3-Methylbutanol	1820.90 ± 120.60 ^bc^	2089.20 ± 162.68 ^ab^	1689.90 ± 70.70 ^c^	2292.50 ± 188.30 ^a^	2000.20 ± 160.01 ^ab^	2325.40 ± 169.76 ^a^	1555.20 ± 146.66 ^c^	1647.40 ± 149.42 ^bc^
1-Pentanol	3.34 ±0.37 ^b^	6.59 ±0.78 ^a^	2.58 ± 0.48 ^b^	3.91 ± 0.82 ^b^	5.67 ± 0.77 ^a^	6.40 ± 0.19 ^a^	3.81 ± 0.16 ^b^	5.45 ± 0.16 ^a^
2-Hexanol	0.52 ± 0.22 ^b^	0.29 ± 0.18 ^b^	12.24 ± 1.37 ^a^	15.42 ± 2.46 ^a^	10.05 ± 1.30 ^b^	76.17 ± 12.29 ^a^	9.57 ± 1.29 ^b^	7.38 ± 1.22 ^b^
1-Hexanol	47.22 ± 5.42 ^b^	63.50 ± 8.71 ^a^	37.67 ± 5.13 ^b^	53.59 ± 4.61 ^ab^	127.26 ± 13.82 ^a^	138.89 ± 14.17 ^a^	71.59 ± 12.15 ^b^	91.76 ± 10.75 ^b^
Furfuryl alcohol	2.68 ± 0.68 ^b^	5.89 ± 2.18 ^ab^	7.11 ± 1.21 ^a^	9.42 ± 1.28 ^a^	7.55 ± 1.23 ^ab^	7.44 ± 1.22 ^b^	10.49 ± 1.31 ^a^	3.16 ± 0.59 ^c^
Benzyl alcohol	14.94 ± 1.45 ^a^	16.01 ± 10.48 ^a^	11.84 ± 1.36 ^a^	15.46 ± 2.46 ^a^	20.24 ± 2.61 ^a^	18.11 ± 2.54 ^a^	14.27 ± 3.43 ^a^	17.60 ± 3.53 ^a^
2-Phenyletanol	67.51 ± 5.03 ^b^	70.44 ± 5.11 ^b^	61.84 ± 16.86 ^b^	218.71 ± 16.56 ^a^	70.63 ±5.12 ^b^	76.08 ± 5.28 ^b^	84.46 ± 7.53 ^b^	114.86 ± 3.45 ^a^
1-Nonanol	40.34 ± 2.21 ^a^	43.37 ± 3.30 ^a^	43.32 ± 3.30 ^a^	46.15 ± 4.38 ^a^	34.87 ± 4.05 ^a^	35.34 ± 4.06 ^a^	37.26 ± 3.12 ^a^	41.83 ± 3.25 ^a^
2-Butanol	0.43 ± 0.08 ^ab^	0.60 ± 0.10 ^a^	0.35 ± 0.08 ^b^	0.34 ± 0.02 ^b^	0.17 ± 0.08 ^b^	0.58 ± 0.06 ^a^	0.17 ± 0.03 ^b^	0.23 ± 0.01 ^b^
1-Octanol	21.29 ± 0.64 ^a^	22.81 ± 0.68 ^a^	16.87 ± 0.51 ^b^	22.03 ± 0.66 ^a^	28.80 ± 3.86 ^a^	25.81 ± 1.77 ^ab^	20.34 ± 1.61 ^b^	25.08 ± 2.75 ^ab^
Acetaldehyde	171.74 ±22.15 ^bc^	278.70 ± 18.36 ^a^	197.09 ± 18.91 ^b^	132.54 ± 23.98 ^c^	216.66 ± 16.5 ^c^	1491.71 ± 15.75 ^a^	343.55 ±19.31 ^b^	349.86 ±14.5 ^b^
Isovaleraldehyde	1.08 ± 0.09 ^a^	1.17 ± 0.09 ^a^	0.57 ± 0.08 ^b^	1.03 ± 0.04 ^a^	1.11 ± 0.09 ^b^	1.75 ± 0.15 ^a^	0.83 ± 0.18 ^b^	1.13 ± 0.13 ^b^
2-Methylbutyraldehyde	1.13 ± 0.15 ^a^	0.62 ± 0.08 ^b^	0.73 ± 0.07 ^b^	1.29 ±0.12 ^a^	1.11 ± 0.13 ^a^	1.33 ± 0.14 ^a^	1.41 ± 0.24 ^a^	2.02 ± 0.16 ^b^
Valeraldehyde	0.33 ± 0.05 ^a^	0.27 ± 0.05 ^a^	0.34 ± 0.07 ^a^	0.22 ± 0.06 ^a^	0.38 ± 0.07 ^bc^	1.15 ± 0.12 ^a^	0.45 ± 0.09 ^b^	0.22 ± 0.05 ^c^
Heksanal	2.98 ± 0.39 ^b^	4.70 ± 0.54 ^a^	2.48 ± 0.27 ^b^	3.50 ± 0.31 ^b^	5.59 ± 0.57 ^a^	7.15 ± 0.81 ^a^	5.71 ± 0.67 ^a^	3.86 ± 0.42 ^b^
Phenylacetaldehyde	1.02 ± 0.13 ^b^	0.79 ± 0.12 ^bc^	0.45 ± 0.06 ^c^	3.16 ± 0.29 ^a^	1.27 ± 0.14 ^b^	3.17 ± 0.24 ^a^	0.66 ± 0.52 ^b^	0.51 ± 0.42 ^b^
Nonanal	19.35 ± 0.88 ^a^	17.78 ± 0.63 ^ab^	17.19 ± 0.82 ^b^	13.02 ± 0.79 ^c^	17.93 ± 2.54 ^c^	43.62 ± 1.31 ^a^	23.67 ± 2.71 ^b^	16.56 ± 1.50 ^c^
Decanal	6.56 ± 0.60 ^a^	7.29 ± 0.62 ^a^	6.86 ± 0.71 ^a^	7.30 ± 0.52 ^a^	4.67 ± 0.54 ^b^	12.38 ± 1.37 ^a^	5.63 ± 0.67 ^b^	3.87 ± 0.42 ^b^
Furfural	10.67 ± 1.32 ^c^	31.92 ± 2.96 ^b^	6.35 ± 0.89 ^c^	51.45 ± 3.54	33.78 ± 2.01 ^c^	113.48 ± 8.40 ^a^	19.44 ± 1.58 ^d^	63.80 ± 4.91 ^b^
Benzaldehyde	21.22 ± 2.64 ^b^	34.58 ± 4.04 ^a^	8.55 ± 1.26 ^c^	10.31 ± 1.11 ^c^	11.89 ± 1.36 ^b^	21.15 ± 1.63 ^a^	7.47 ± 0.98 ^c^	9.88 ± 1.52 ^bc^
Isobutyraldehyde	43.48 ± 2.30 ^a^	n.d.	44.21 ± 3.33 ^a^	43.28 ± 3.30 ^a^	41.48 ± 3.24 ^a^	39.86 ± 3.20 ^a^	43.41 ± 3. ^a^ 30	48.91 ± 4.47 ^a^
Trans-2-heptanal	20.98 ± 3.63 ^a^	18.79 ± 2.56 ^a^	18.57 ± 2.56 ^a^	20.18 ± 2.61 ^a^	19.91 ± 1.60 ^a^	20.05 ± 2.18 ^a^	19.01 ± 1.57 ^a^	19.63 ± 1.59 ^a^
3-Etoksypropionaldehyde	16.61 ± 1.50 ^a^	16.62 ± 1.23 ^a^	16.38 ± 1.49 ^a^	n.d.	n.d.	19.82 ± 1.60 ^a^	18.47 ± 1.55 ^a^	17.09 ± 1.51 ^a^
**Compound** **[mg/L alcohol 100% v/v]**	**Delikates**				**Kronselska**			
	**Raw pulp**	**Deacidification**	**Pasteurization**	**Raw pulp**	**Deacidification**	**Pasteurization**	**Raw pulp**	**Deacidification**
2-Propanone	n.d.	8.53 ± 0.76 ^a^	4.30 ± 0.52 ^b^	5.33 ± 0.76 ^b^	4.32 ± 0.63 ^b^	9.74 ± 0.89 ^a^	4.41 ± 0.33 ^b^	3.48 ± 0.67 ^b^
2,3-Butanedione	3.81 ± 0.51 ^a^	0.48 ± 0.16 ^c^	0.22 ± 0.09 ^c^	2.03 ± 0.36 ^b^	5.93 ± 0.38 ^b^	18.28 ± 1.55 ^a^	0.75 ± 0.08 ^c^	2.15 ± 0.46 ^c^
Acetaldehyde diethyl acetal	1131.90 ± 33.96 ^c^	1627.30 ± 48.82 ^b^	1856.1 ± 55.68 ^a^	823.4 ± 24.7 ^d^	1616.2 ^b^ ± 48.5	1208.9 ^c^ ± 36.3	2419.4 ^a^ ± 72.6	583.6 ^d^ ± 17.5
2,3-Pentanodione	10.39 ^ab^ ±1.31	8.28 ^b^ ± 0.95	11.87 ^a^ ± 1.36	10.01 ^ab^ ± 1.30	17.37 ± 2.52 ^a^	12.96 ± 1.39 ^a^	16.35 ± 1.49 ^a^	22.50 ± 1.68 ^b^
3-Octanone	37.13 ± 15.11 ^ab^	50.27 ± 4.51 ^a^	22.59 ± 3.68 ^b^	27.16 ± 3.81 ^b^	28.63 ± 3.86 ^ab^	37.06 ± 5.11 ^a^	24.55 ± 3.74 ^b^	26.76 ± 3.80 ^ab^
Ethyl formate	1.37 ± 0.07 ^c^	4.76 ± 0.24 ^a^	0.42 ± 0.02 ^d^	2.39 ± 0.12 ^b^	3.83 ± 0.19 ^a^	1.83 ± 0.39 ^b^	1.49 ± 0.27 ^b^	1.42 ± 0.27 ^b^
Ethyl acetate	339.80 ± 41.99 ^b^	712.00 ± 59.60 ^a^	367.30 ± 40.62 ^b^	314.70 ± 25.89 ^b^	383.00 ±34.15 ^b^	869.20 ± 34.56 ^a^	355.80 ± 39.79 ^b^	301.50 ± 65.08 ^b^
Isobutyl acetate	2.03 ± 0.29 ^a^	1.36 ± 0.17 ^b^	0.75 ± 0.08 ^c^	0.40 ± 0.09 ^c^	1.36 ± 0.27 ^b^	9.61 ± 0.48 ^a^	0.88 ± 0.09 ^b^	1.51 ± 0.28 ^b^
Butyl acetate	1.26 ± 0.26 ^b^	0.78 ± 0.19 ^bc^	1.91 ± 0.35 ^a^	0.53 ± 0.12 ^c^	6.71 ± 0.94 ^b^	8.43 ± 1.42 ^b^	8.01 ± 0.40 ^b^	13.56 ± 2.68 ^a^
3-Methylbutyl acetate	14.32 ± 1.72 ^a^	14.15 ± 0.71 ^ab^	9.73 ± 2.49 ^bc^	9.34 ± 1.47 ^c^	14.31 ± 1.72 ^c^	37.01 ± 2.85 ^b^	38.04 ± 1.90 ^b^	76.45 ± 3.82 ^a^
2-Methylbutyl acetate	1.22 ± 0.16 ^a^	1.16 ± 0.16 ^a^	1.42 ± 0.17 ^a^	0.64 ± 0.09 ^b^	1.98 ± 0.15 ^c^	9.44 ± 0.47 ^a^	1.89 ± 0.19 ^c^	3.08 ± 0.15 ^b^
Hexyl acetate	0.20 ± 0.01 ^b^	0.20 ± 0.01 ^b^	0.60 ± 0.03 ^a^	0.12 ± 0.01 ^c^	0.98 ± 0.049 ^c^	2.33 ± 0.12 ^b^	3.28 ± 0.16 ^a^	3.24 ± 0.16 ^a^
2-Phenylethyl acetate	2.40 ± 0.35 ^b^	2.35 ± 0.32 ^b^	3.90 ± 0.27 ^a^	2.96 ± 0.35 ^b^	7.72 ± 0.69 ^c^	8.89 ± 1.94 ^bc^	13.85 ± 2.69 ^ab^	18.31 ± 2.12 ^a^
Methyl acetate	2.28 ± 0.11 ^c^	30.85 ± 1.54 ^a^	9.12 ± 0.46 ^b^	30.32 ± 1.52 ^a^	12.58 ±0.63 ^c^	428.23 ±21.41 ^a^	n.d.	230.95 ±11.54 ^b^
Ethyl propanoate	1.93 ± 0.35 ^b^	1.52 ± 0.18 ^b^	2.76 ± 0.14 ^a^	2.60 ± 0.13 ^a^	8.75 ± 0.94 ^b^	5.13 ± 0.26 ^c^	7.27 ± 0.86 ^bc^	11.65 ± 1.58 ^a^
Isoamyl propanoate	0.08 ± 0.01 ^a^	0.06 ± 0.01 ^a^	0.08 ± 0.02 ^a^	0.07 ± 0.01 ^a^	0.14 ± 0.02 ^a^	0.18 ± 0.04 ^a^	0.15 ± 0.01 ^a^	0.21 ± 0.05 ^a^
Ethyl isobutyrate	0.41 ± 0.11 ^a^	0.48 ± 0.12 ^a^	0.30 ± 0.07 ^ab^	0.14 ± 0.01 ^b^	0.19 ± 0.03 ^b^	4.89 ± 0.24 ^a^	0.29 ± 0.07 ^b^	0.39 ± 0.09 ^b^
Ethyl butyrate	1.00 ± 0.08 ^a^	1.14 ± 0.06 ^a^	0.78 ± 0.07 ^b^	0.66 ± 0.09 ^b^	0.86 ± 0.14 ^b^	2.85 ± 0.14 ^a^	0.64 ± 0.13 ^b^	0.54 ± 0.11 ^b^
Ethyl 2-methylbutyrate	0.26 ± 0.05 ^a^	0.33 ± 0.05 ^a^	0.06 ± 0.01 ^b^	0.09 ± 0.03 ^b^	0.13 ± 0.04 ^b^	0.70 ± 0.14 ^a^	0.03 ± 0.01 ^b^	0.05 ± 0.02 ^b^
Ethyl 3-methylbutyrate	0.28 ± 0.01 ^b^	0.54 ± 0.03 ^a^	0.13 ± 0.04 ^c^	0.07 ± 0.03 ^c^	0.05 ± 0.02 ^b^	2.50 ± 0.13 ^a^	0.07 ± 0.02 ^b^	0.05 ± 0.01 ^b^
Ethyl hexanoate	3.62 ± 0.88 ^ab^	4.93 ± 0.75 ^a^	1.92 ± 0.10 ^c^	2.21 ± 0.27 ^bc^	2.74 ± 0.36 ^b^	7.85 ± 0.89 ^a^	2.10 ± 0.11 ^b^	1.61 ± 0.39 ^b^
Ethyl heptanoate	0.18 ± 0.02 ^a^	0.22 ± 0.05 ^a^	0.13 ± 0.06 ^a^	0.10 ± 0.05 ^a^	0.11 ± 0.01 ^c^	0.26 ± 0.04 ^a^	0.19 ± 0.03 ^b^	0.12 ± 0.01 ^c^
Ethyl octanoate	9.42 ± 1.47 ^a^	11.44 ± 2.57 ^a^	10.32 ± 1.52 ^a^	10.91 ± 1.55 ^a^	6.90 ± 1.35 ^a^	8.20 ± 2.91 ^a^	8.82 ± 2.44 ^a^	6.45 ± 2.32 ^a^
Ethyl nonanoate	0.31 ± 0.02 ^b^	0.61 ± 0.03 ^a^	0.27 ± 0.04 ^b^	0.31 ± 0.03 ^b^	0.26 ± 0.05 ^b^	0.86 ± 0.04 ^a^	0.33 ± 0.04 ^b^	0.33 ± 0.02 ^b^
Ethyl decanoate	12.30 ± 1.62 ^b^	13.46 ± 1.67 ^ab^	15.98 ± 2.80 ^ab^	19.26 ± 2.96 ^a^	8.27 ±0.41 ^c^	27.69 ± 2.38 ^a^	15.71 ± 2.79 ^b^	12.51 ± 1.63 ^bc^
Ethyl lactate	12.23 ± 1.61 ^a^	10.13 ± 1.36 ^a^	0.24 ± 0.03 ^b^	n.d.	49.83 ± 3.49 ^a^	46.42 ± 2.82 ^a^	n.d.	n.d.
Ethyl benzoate	8.96 ± 0.84 ^a^	9.52 ± 0.45 ^a^	10.41 ± 1.52 ^a^	11.90 ± 2.60 ^a^	5.98 ± 0.30 ^c^	2.14 ± 0.11 ^c^	10.93 ± 1.55 ^a^	8.71 ± 1.44 ^ab^
Ethyl tetradecanoate	6.59 ± 0.83 ^a^	4.96 ± 0.72 ^a^	5.38 ± 0.27 ^a^	5.07 ± 0.75 ^a^	9.19 ± 1.46 ^b^	18.68 ± 1.93 ^a^	7.40 ± 1.36 ^b^	6.59 ± 0.33 ^b^

Results expressed as mean values ± SD (n = 3); means in a row with different superscript letters (^a–d^), separately for each apple cultivar, are significantly different as analyzed by one-way ANOVA and Tukey’s post hoc. test; n.d.—not detected.

## Data Availability

Data is contained within the article.

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
