# Peer review of "Effect of Apple Cultivar and Selected Technological Treatments on the Quality of Apple Distillate"

_foods, 2023, doi:10.3390/foods12244494_

Round 1

Reviewer 1 Report

Comments and Suggestions for Authors

I reviewed the manuscript titled “Effect of apple cultivar and selected technological treatments on the quality of apple distillate. The manuscript has novelty and contributes to the field.

Remove the period after the end of title

Abstract

Authors must restructure the abstract. Introduce the background of the study; research objectives; results; conclusions and recommendations

As such, authors wrote the abstract in paragraphs, which is mostly unexpected in the abstract.

Keywords: the keyword “apple cultivars” must be introduced

Line 34: France is word famous.. is it world famous?

Introduction is not clear

Authors should focus on background; research gap with clear research objectives

For example, the objectives of the study must be revised. As such, it looks like a research progress report. I suggest authors revise it accordingly.

Section 2.4.: this section is not clear. Authors must provide detailed methodology with appropriate citation

Provide a reference citation  for Free hydrocyanic acid (HCN) content in the tested distillates

Provide a citation for “Samples of apple distillates diluted to 35% v/v were subjected”

Table 1: authors should represent mean ± SD instead of separate presentation. For example, 15 ± 0.50

Table 2: follow the same as Table 1 and other Tables

For Tables, Since authors showing the differences using a or b etc, I feel it is redundant to show its P value.

Authors should represent mean ± standard deviation (superscript difference). Placing standard deviation separately along with individual p value is not recommended, at least for food science research

Quality of the Figures must be improved. For example, X and Y axis lines must be drawn clearly.

Data below the figure is not necessary. Authors must denote statistical differences above the error bar, just like Figure 2.

Discussion of results is sufficient

References must be cross-checked. For example, ref 71, remove bracket “)”

Reviewer 2 Report

Comments and Suggestions for Authors

Review: „Effect of apple cultivar and selected technological treatments on the quality of apple distillate”

In their manuscript, the authors discuss, as the title suggests, the quality of apple distillates from four different apple varieties and different processing methods. The authors' statements are based on investigations of the fruit flesh´s fermentation, the use of various processing methods such as deacidification or pasteurization, or the concentration of various ingredients.

The manuscript is well written and requires only minor English checks. I recommend major revisions.

Remarks:

Chapter 2.4.: Reorganizes that chapter by sub-chapters broken down by topic.

Statistics: When you are using an ANOVA, you have a a-d notation. Please explain.

Line 210: (IV) Is it an oxidation step of CO2?

Chapter 3.2: The discussion could be more detailed. Are there already studies on this in the current literature?

Line 313: Fructose is high. Are there any explanations? Growth conditions and so on…

Page numbers: Your page numbers are a little messed up. Please correct.

Fig 3 and 4: Is there a reason for the cut error bars above a value of 5. Maybe you have one, but it is not very well explained.

Comments on the Quality of English Language

Although I´m not a mother tongue, I think the manuscript only needs minor English checks.

Round 2

Reviewer 1 Report

Comments and Suggestions for Authors

Authors improved the quality of manuscript. However, few points needs to be addressed.

All Tables: Please revise "meansuperscript ± standard deviation" to mean ± standard deviation superscript 

Also, use X and Y axis outer tick marks for Figures. 

Increase the thickness of X and Y axis 

Author Response

Dear Reviewer,

we would like to inform that all sugested changes in the manuscript have been intoduced, i.e. presentation of results in all Tables have been changed from
"mean superscript ± standard deviation" to mean ± standard deviation superscript 

Also, we used X and Y axis outer tick marks for Figures, and the thickness of X and Y axis have been increased.

We hope that all revisions made will satisfy you, and you will find our manuscript acceptable for publication in the journal Foods. 

Reviewer 2 Report

Comments and Suggestions for Authors

Dear editor, dear authors.

You improved your manuscript a lot. With this in mind, I would like to thank you for incorporating my suggestions for improvement. I therefore have no further reservations against publishing the manuscript in "foods". However, when reviewing it again, I found a few deviations from the foods template (no author affiliations, line numbers and some tables are on top of each other and are illegible). Please correct. After all, I recommend publishing the manuscript in “foods” in the present form.

Comments on the Quality of English Language

Minor editing of English language required

Author Response

Dear Reviewer,  Thank you very much for recommending to publish the manuscript in the journal Foods  in the present form.